# Masked Audio Generation using a Single Non-Autoregressive Transformer

[*]**Alon Ziv**[1,3]**, Itai Gat**[1]**, Gael Le Lan**[1]**, Tal Remez**[1]**, Felix Kreuk**[1]**, Alexandre Défossez**[2]
**Jade Copet**[1]**, Gabriel Synnaeve**[1]**, Yossi Adi**[1,3]

[1]FAIR Team, Meta
[2]Kyutai
[3]The Hebrew University of Jerusalem
`alonzi@cs.huji.ac.il`

## Abstract

We introduce MAGNeT, a masked generative sequence modeling method that operates directly over several streams of audio tokens. Unlike prior work, MAGNeT is comprised of a single-stage, non-autoregressive transformer. During training, we predict spans of masked tokens obtained from a masking scheduler, while during inference we gradually construct the output sequence using several decoding steps. To further enhance the quality of the generated audio, we introduce a novel rescoring method in which, we leverage an external pre-trained model to rescore and rank predictions from MAGNeT, which will be then used for later decoding steps. Lastly, we explore a hybrid version of MAGNeT, in which we fuse between autoregressive and non-autoregressive models to generate the first few seconds in an autoregressive manner while the rest of the sequence is being decoded in parallel. We demonstrate the efficiency of MAGNeT for the task of text-to-music and text-to-audio generation and conduct an extensive empirical evaluation, considering both objective metrics and human studies. The proposed approach is comparable to the evaluated baselines, while being significantly faster (x7 faster than the autoregressive baseline). Through ablation studies and analysis, we shed light on the importance of each of the components comprising MAGNeT, together with pointing to the trade-offs between autoregressive and non-autoregressive modeling, considering latency, throughput, and generation quality. Samples are available on our demo page `https://pages.cs.huji.ac.il/adiyoss-lab/MAGNeT`

## 1 Introduction

Recent developments in self-supervised representation learning (Hsu et al., 2021; Défossez et al., 2022), sequence modeling (Touvron et al., 2023; Rozière et al., 2023), and audio synthesis (Lee et al., 2022; Polyak et al., 2021) allow a great leap in performance when considering high quality conditional audio generation. The prominent approach, in recent years, is to represent the audio signals as a compressed representation, either discrete or continuous, and apply a generative model on top of it (Lakhotia et al., 2021; Kharitonov et al., 2022; Borsos et al., 2023a; Kreuk et al., 2022a; Copet et al., 2023; Lam et al., 2023; Agostinelli et al., 2023; Gat et al., 2023; Sheffer & Adi, 2023; Maimon & Adi, 2022; Schneider et al., 2023; Huang et al., 2023b; Liu et al., 2023a; Li et al., 2023; Liu et al., 2023b). Recently, Défossez et al. (2022); Zeghidour et al. (2021) proposed to apply a VQ-VAE directly on the raw waveform using residual vector quantization to obtain a multi-stream discrete representation of the audio signal. Later on, Kreuk et al. (2022a); Wang et al. (2023); Zhang et al. (2023); Copet et al. (2023); Kreuk et al. (2022b) presented a conditional language modeling on such audio signals representations. In parallel, Schneider et al. (2023); Huang et al. (2023b); Liu et al. (2023a) proposed training a conditional diffusion-based generative model operating on learned continuous representations of the audio signal obtained from a pre-trained auto-encoder model.

---

[*]Work was done as part of Alon's internship at FAIR.

Overall, the family of generative models explores in prior work can be roughly divided into two: (i) autoregressive (AR) in the form of language models (LMs), usually operating on discrete audio representations; and (ii) diffusion-based models usually operating on continuous latent representations of the audio signal. Although providing impressive generation results, these approaches have several main drawbacks. Due to its autoregressive nature, following the LM approach yields relatively high inference time which turns into high latency values, hence making it less appalling for interactive applications such as music generation and editing under Digital Audio Workstations (DAW). On the other hand, diffusion models perform parallel decoding, however, to reach high-quality music samples recent studies report using a few hundred diffusion decoding steps (Huang et al., 2023a; Liu et al., 2023b). Moreover, diffusion models struggle with generating long-form sequences. Recent studies present results for either 10-second generations (Liu et al., 2023b; Li et al., 2023; Yang et al., 2022) or models that operate in low resolution followed by a cascade of super-resolution models to reach 30-second segments (Huang et al., 2023a).

In this work, we present MAGNET, a short for **M**asked **A**udio **G**eneration using **N**on-autoregressive **T**ransformers. MAGNET is a novel masked generative sequence modeling operating on a multi-stream representation of an audio signal. The proposed approach comprised of a single transformer model, working in a non-autoregressive fashion. During training, we first sample a masking rate from the masking scheduler, then, we mask and predict spans of input tokens conditioned on unmasked ones. During inference, we gradually build the output audio sequence using several decoding steps. We start from a fully masked sequence and at each iteration step, we fix the most probable token spans, i.e., the spans that got the top confidence score. To further enhance the quality of the generated audio, we introduce a novel rescoring method. In which, we leverage an external pre-trained model to rescore and rank predictions from MAGNET. Lastly, we explore a Hybrid version of MAGNET, in which we fuse autoregressive and non-autoregressive models. The hybrid-MAGNET generates the beginning of the tokens sequence in an autoregressive manner while the rest of the sequence is being decoded in parallel, similarly to the original MAGNET. A visual description of the inference of the proposed method can be seen in Fig. 1.

Similar non-autoregressive modeling was previously proposed by Ghazvininejad et al. (2019) for machine translation, Chang et al. (2022) for class-conditional image generation and editing, and Chang et al. (2023) for image generation guided by rich textual description followed by a super-resolution component. Borsos et al. (2023b) recently proposed SoundStorm, a non-autoregressive method for the task of text-to-speech and dialogue synthesis. SoundStorm is conditioned on "semantic" tokens obtained from an autoregressive model. Unlike, SoundStorm, MAGNET is composed of a single non-autoregressive model and was evaluated on music and audio generation which, unlike speech, leverages the full frequency spectrum of the signal.

We evaluate the proposed approach considering both text-to-music and text-to-audio generation. We report objective metrics together with a human study and show the proposed approach achieves comparable results to the evaluated baselines while having significantly reduced latency (x7 faster than the autoregressive-based method). We further present an analysis of the proposed method considering latency, throughput, and generation quality. We present the trade-offs between the two when considering autoregressive and non-autoregressive models. Lastly, we provide an ablation study that sheds light on the contribution of each component of the proposed approach to the performance.

**Our contributions:** (i) We present a novel non-autoregressive model for the task of audio modeling and generation, denoted as MAGNET. The proposed method is able to generate relatively long sequences (30 seconds long), using a single model. The proposed approach has a significantly faster inference time while reaching comparable results to the autoregressive alternative; (ii) We leverage an external pre-trained model during inference to improve generation quality via a rescoring method; and (iii) We show how the proposed method can be combined with autoregressive modeling to reach a single model that performs joint optimization, denoted as Hybrid-MAGNET.

## 2 BACKGROUND

**Audio representation.** Modern audio generative models mostly operate on a latent representation of the audio, commonly obtained from a compression model (Borsos et al., 2023a; Kreuk et al., 2022a; Yang et al., 2022). Compression models such as Zeghidour et al. (2021) employ Residual Vector Quantization (RVQ) which results in several parallel streams. Under this setting, each stream

is comprised of discrete tokens originating from different learned codebooks. Prior work, proposed several modeling strategies to handle this issue (Kharitonov et al., 2022; Wang et al., 2023).

Specifically, Défossez et al. (2022) introduced EnCodec , a convolutional auto-encoder with a latent space quantized using Residual Vector Quantization (RVQ) (Zeghidour et al., 2021), and an adversarial reconstruction loss. Given a reference audio signal $x \in \mathbb{R}^{d \cdot f_s}$ with $d$ the audio duration and $f_s$ the sample rate, EnCodec first encodes it into a continuous tensor with a frame rate $f_r \ll f_s$. Then, this representation is quantized into $z \in \{1, \ldots, N\}^{K \times d \cdot f_r}$, with $K$ being the number of codebooks used in RVQ and $N$ being the codebook size. Notice, after quantization we are left with $K$ discrete token sequences, each of length $T = d \cdot f_r$, representing the audio signal. In RVQ, each quantizer encodes the quantization error left by the previous quantizer, thus quantized values for different codebooks are in general dependent, where the first codebook is the most important one.

**Audio generative modeling.** Given a discrete representation of the audio signal, $z$, our goal is to model the conditional joint probability distribution $p_\theta(z|y)$, where $y$ is a semantic representation of the condition. Under the autoregressive setup we usually follow the chain rule of probability, thus the joint probability of a sequence can be computed as a product of its conditional probabilities:

$$p_\theta(z_1, \ldots, z_n | y) = \prod_{i=1}^{n} p_\theta(z_i | z_{i-1}, \ldots, z_1, y). \tag{1}$$

The above probability chain rule can be thought of as a masking strategy, where, in each time step $i$, we predict the probability of the $i$-th token, given its past tokens, while we mask future tokens. For that, we define a masking function $m(i)$, that mask out all tokens larger than $i$, which results in:

$$p_\theta(z_1, \ldots, z_n | y) = \prod_{i=1}^{n} p_\theta(z_i | (1 - m(i)) \odot z, y), \tag{2}$$

where each element in $m(i) = [m_1(i), \ldots, m_T(i)]$ is defined as $m_j(i) = \mathbb{1}[j \geq i]$. Notice, Eq. (2) does not hold for any masking strategy. One should pick a masking strategy that satisfies the probability chain rule.

Extending Eq. (2) to the non-autoregressive setup can be done by modifying the masking strategy and the decomposition of the joint probability to predict an arbitrary subset of tokens given the unmasked ones using several decoding steps. Let us formally define the masking strategy as follows,

$$m_j(i) \sim \mathbb{1}[j \in \mathcal{M}_i] \quad \text{where} \quad \mathcal{M}_i \sim \mathcal{U}(\{\mathcal{A} \subseteq \mathcal{M}_{i-1} : |\mathcal{A}| = \gamma(i; s) \cdot T\}), \tag{3}$$

and $\gamma$ is the masking scheduler, with $s$ decoding steps, defined as $\gamma(i; s) = \cos(\frac{\pi(i-1)}{2s})$ and $\mathcal{M}_0 = \{1, \ldots, T\}$. In other words, at each time step $i$ we mask a subset of $\gamma(i; s) \cdot T$ tokens sampled from the masked set at the previous time step. Thus the modified version of Eq. (2) is,

$$p_\theta(z_1, \ldots, z_n | y) = \prod_{i=1}^{s} p_\theta(m(i) \odot z | (1 - m(i)) \odot z, y). \tag{4}$$

In practice, during training, a decoding time step $i \in [1, s]$ and the tokens to be masked from $\mathcal{M}_0$ are randomly sampled. The tokens at indices $t \in \mathcal{M}_i$ are then replaced by a special mask token, and the model is trained to predict the target tokens at the masked positions $\mathcal{M}_i$ given the unmasked tokens. This modeling paradigm was previously explored by Ghazvininejad et al. (2019); Chang et al. (2022; 2023); Borsos et al. (2023b).

Recall, the audio representation is composed of multi-stream sequences created by RVQ. In which, the first codebook encodes the coarse information of the signal while later codebooks encode the quantization error to refine the generation quality. To handle that, Borsos et al. (2023b) proposed to predict tokens from codebook $k$ given its preceding codebooks. During training, a codebook level $k$, is being uniformly sampled from $\{1, \ldots, K\}$. Then, we mask and predict the tokens of the $k$-th codebook given previous levels via teacher forcing. At inference, we sequentially generate the token streams, where each codebook is being generated conditioned on previously generated codebooks.

## 3 METHOD

Following the approach presented in the previous section solely does not lead to high-quality audio generation. We hypothesize this is due to three factors: (i) The masking strategy operates over

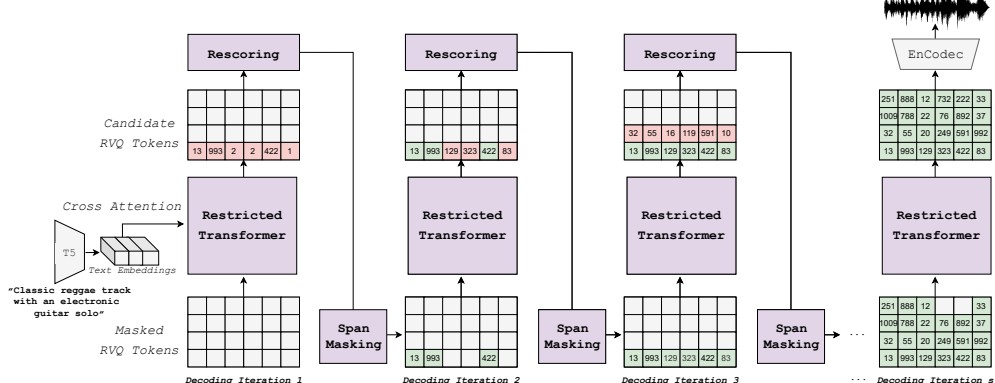

Figure 1: Inference of MAGNET model. During each iteration, we mask a subset of token spans (starting from a fully masked sequence). Next, we rescore the tokens based on an external pre-trained model. Finally, we select the token spans to be re-masked for the next decoding iteration.

individual tokens that share information with adjacent tokens. Hence, allowing the model to "cheat" during tokens prediction while being trained using teacher forcing; (ii) The temporal context of the codebooks at levels greater than one, is generally local and influenced by a small set of neighboring tokens. This affects model optimization; (iii) Sampling from the model at different decoding steps requires different levels of diversity with respect to the condition. Also sampling can be combined with external scoring models.

In this section, we present MAGNET in details. MAGNET consists of a non-autoregressive audio-based generative masked language model, conditioned on a semantic representation of the condition, operating on several streams of discrete audio tokens obtained from EnCodec . We follow a similar modeling strategy as presented in Section 2 while introducing core modeling modifications consisting of masking strategy, restricted context, sampling mechanism, and model rescoring.

## 3.1 MASKING STRATEGY

Adjacent audio tokens often share information due to the receptive field of the audio encoder. Hence, we use spans of tokens as the atomic building block of our masking scheme, rather than individual ones as done in prior work. We evaluated various span lengths $l$ between 20ms to 200ms and found a 60ms span length to give the best overall performance (see Section 5.3 for detailed results). We sample a masking rate $\gamma(i)$, from the scheduler, and compute the average amount of spans to be masked accordingly. As spans may overlap, this process requires a careful design. We select the number of spans $u$, that satisfies $1 - \binom{T-l}{u}/\binom{T}{u} \approx \gamma(i)$, where $l$ is the span length. The above expression is the expected masking rate over all possible placements of $u$ spans of length $l$ over the sequence. Full derivation can be found in the Appendix C. During inference time, we follow a similar strategy, in which we re-mask the least probable spans instead of individual tokens as done in prior work. We consider the span's probability as the token with the maximal probability. For computational efficiency, we use non-overlapping spans.

## 3.2 RESTRICTED CONTEXT

Recall, the used audio tokenizer is based on RVQ, where each quantizer encodes the quantization error left by the previous quantizer. Thus quantized codebooks later than the first one heavily depend on previous codebooks rather than surrounding tokens. To leverage that we analyze the used EnCodec and restrict the context of the codebooks accordingly.

Specifically, the audio encoder consists of a multi-layer convolutional network and a final LSTM block. Analyzing the receptive field for the used EnCodec shows that the receptive field of the convolutional network is $\sim 160$ms, while the effective receptive field when including the LSTM block is $\sim 180$ms. We empirically estimate the receptive field of the model, using a shifted impulse function over time while measuring the magnitude of the encoded vector in the middle of the sequence.

Fig. 3 in Appendix G depicts such process. Notice, although theoretically, the LSTM has an infinite memory, practically we observe it is bounded.

We utilize this observation to improve model optimization, by restricting the self-attention of code-books greater than 1, to attend only on tokens at a temporal distance smaller than $\sim 200$ms. Similar ideas were proposed in the context of language modeling by Rae & Razavi (2020); Roy et al. (2021). We depict the used attention map for the restricted context in Fig. 8.

## 3.3 MODEL INFERENCE

**Sampling** as described in Eq. (3) uses a uniform sampling to choose spans from the previously set of masked spans. In practice, we use the model confidence at the $i$-th iteration as a scoring function to rank all possible spans and choose the least probable spans to be masked accordingly. However, the scoring function does not have to be part of the generative model.

A common practice in Automatic Speech Recognition (ASR) decoding, is to generate a set of different hypotheses from one model and rescore them using another model (Benesty et al., 2008; Likhomanenko et al., 2020). Inspired by the ASR rescoring method, we propose a novel strategy in which at iteration $i$ we generate a candidate token sequence using MAGNET. Then, we feed it to an external model and get a new set of probabilities for each of the token spans. Lastly, we use a convex combination of both probabilities (the one emitted by MAGNET and the one obtained from the rescorer model), to sample from:

$$p(z|y) = w \cdot p_\theta(z|y) + (1 - w) \cdot p_{\text{rescorer}}(z|y). \tag{5}$$

In this work, we use MUSICGEN and AudioGen as our rescorering models (in a non-autoregressive manner). The proposed rescoring method is generic and is not tied to any specific rescoring model. Following the proposed approach improves the generated audio quality and stabilizes inference. A pseudo-code of our entire decoding algorithm is described in Fig. 4, Appendix D.

**Classifier-free guidance annealing.** Token prediction is done using a Classifier-Free Guidance (CFG) (Ho & Salimans, 2022). During training, we optimize the model both conditionally and unconditionally, while at inference time we sample from a distribution obtained by a linear combination of the conditional and unconditional probabilities.

While prior work (Copet et al., 2023; Kreuk et al., 2022a) used a fixed guidance coefficient, $\lambda > 1$, we instead use a CFG annealing mechanism controlled by the masking schedule $\gamma$. As the masking rate $\gamma(i)$ decreases, the guidance coefficient is annealed during the iterative decoding process. The motivation behind this approach is to gradually reduce text adherence and guide the generation process toward the already fixed tokens. Intuitively, this transforms the sampling process from textually guided to contextual infilling. Formally, we use a CFG coefficient of

$$\lambda(i) = \gamma(i) \cdot \lambda_0 + (1 - \gamma(i)) \cdot \lambda_1, \tag{6}$$

where $\lambda_0$ and $\lambda_1$ are the initial and final guidance coefficients respectively. This approach was also found to be beneficial in 3D shape generation (Sanghi et al., 2023).

## 4 EXPERIMENTAL SETUP

**Implementation details.** We evaluate MAGNET on the task of text-to-music generation and text-to-audio generation. We use the exact same training data as using by Copet et al. (2023) for music generation and by Kreuk et al. (2022a) for audio generation. A detailed description of the used dataset can be found on Appendix A.2. We additionally provide a detailed description about the datasets used to train the evaluated baselines in Table 4.

Under all setups, we use the official EnCodec model as was published by Copet et al. (2023); Kreuk et al. (2022a)[1]. The model gets as input an audio segment and outputs a 50 Hz discrete representation. We use four codebooks, where each has a codebook size of 2048. We perform the same text preprocessing as proposed by Copet et al. (2023); Kreuk et al. (2022a). We use a pre-trained T5 Raffel et al. (2020) model to extract semantic representation from the text description and use it as model conditioning.

---

[1]https://github.com/facebookresearch/audiocraft

Table 1: Comparison to Text-to-Music Baselines. The Mousai and MusicGen models were retrained on the same dataset, while for MusicLM we use the public API for human studies. We report the original FAD for AudioLDM2, and MusicLM. For human studies, we report mean and CI95.

| MODEL | $FAD_{vgg}$ ↓ | KL ↓ | $CLAP_{scr}$ ↑ | OVL. ↑ | REL. ↑ | # STEPS | LATENCY (S) |
|---|---|---|---|---|---|---|---|
| Reference | - | - | - | $92.69_{\pm0.89}$ | $93.97_{\pm0.82}$ | - | - |
| Mousai | 7.5 | 1.59 | 0.23 | $73.97_{\pm1.93}$ | $74.12_{\pm1.43}$ | 200 | 44.0 |
| MusicLM | 4.0 | - | - | $84.03_{\pm1.28}$ | $85.57_{\pm1.12}$ | - | - |
| AudioLDM 2 | 3.1 | 1.20 | 0.31 | $77.69_{\pm1.93}$ | $82.41_{\pm1.36}$ | 208 | 18.1 |
| MUSICGEN-small | 3.1 | 1.29 | 0.31 | $84.68_{\pm1.45}$ | $83.89_{\pm1.01}$ | 1500 | 17.6 |
| MUSICGEN-large | 3.4 | 1.23 | 0.32 | $85.65_{\pm1.51}$ | $84.12_{\pm1.12}$ | 1500 | 41.3 |
| MAGNET-small | 3.3 | 1.12 | 0.31 | $81.67_{\pm1.72}$ | $83.21_{\pm1.17}$ | 180 | 4.0 |
| MAGNET-large | 4.0 | 1.15 | 0.29 | $84.26_{\pm1.43}$ | $84.21_{\pm1.34}$ | 180 | 12.6 |

We train non-autoregressive transformer models using 300M (MAGNET-small) and $1.5B$ (MAGNET-large) parameters. We train models using 30-second audio crops sampled at random from the full track. We train the models for 1M steps with the AdamW optimizer (Loshchilov & Hutter, 2017), a batch size of 192 examples, $\beta_1 = 0.9$, $\beta_2 = 0.95$, a decoupled weight decay of 0.1 and gradient clipping of 1.0. We further rely on D-Adaptation-based automatic step-sizes (Defazio & Mishchenko, 2023). We use a cosine learning rate schedule with a warmup of 4K steps. Additionally, we use an exponential moving average with a decay of 0.99. We train the models using respectively 32 GPUs for small and 64 GPUs for large models, with float16 precision. For computational efficiency, we train 10-second generation models with a batch size of 576 examples for all ablation studies. Finally, for inference, we employ nucleus sampling (Holtzman et al., 2020) with top-p 0.9, and a temperature of 3.0 that is linearly annealed to zero during decoding iterations. We use CFG with a condition dropout of 0.3 at training, and a guidance coefficient 10.0 annealed to 1.0.

**Evaluation metrics.** We evaluate the proposed method using the same setup as proposed by Copet et al. (2023); Kreuk et al. (2022a), which consists of both objective and subjective metrics. For the objective metrics, we use: the Fréchet Audio Distance (FAD), the Kullback-Leiber Divergence (KL), and the CLAP score (CLAP). We report the FAD (Kilgour et al., 2018) using the official implementation in Tensorflow with the VGGish model [2]. Following Kreuk et al. (2022a), we use a state-of-the-art audio classifier (Koutini et al., 2021) to compute the KL-divergence over the probabilities of the labels between the original and the generated audio. We also report the CLAP score (Wu et al., 2023; Huang et al., 2023b) between the track description and the generated audio to quantify audio-text alignment, using the official CLAP model [3].

For the human studies, we follow the same setup as in Kreuk et al. (2022a). We ask human raters to evaluate two aspects of the audio samples (i) overall quality (OVL), and (ii) relevance to the text input (REL). For the overall quality test, raters were asked to rate the perceptual quality of the provided samples in a range of 1 to 100. For the text relevance test, raters were asked to rate the match between audio and text on a scale of 1 to 100. Raters were recruited using the Amazon Mechanical Turk platform. We evaluate randomly sampled files, where each sample was evaluated by at least 5 raters. We use the CrowdMOS package[4] to filter noisy annotations and outliers. We remove annotators who did not listen to the full recordings, annotators who rate the reference recordings less than 85, and the rest of the recommended recipes from CrowdMOS (Ribeiro et al., 2011).

## 5 RESULTS

### 5.1 TEXT-TO-MUSIC GENERATION

We compare MAGNET to Mousai (Schneider et al., 2023), MusicGen Copet et al. (2023), AudioLDM2 Liu et al. (2023b) [5], and MusicLM (Agostinelli et al., 2023). We train Mousai using our dataset using the open source implementation provided by the authors[6].

---

[2] github.com/google-research/google-research/tree/master/frechet_audio_distance

[3] https://github.com/LAION-AI/CLAP

[4] http://www.crowdmos.org/download/

[5] huggingface.co/spaces/haoheliu/audioldm2-text2audio-text2music (September 2023)

[6] Implementation from github.com/archinetai/audio-diffusion-pytorch (March 2023)

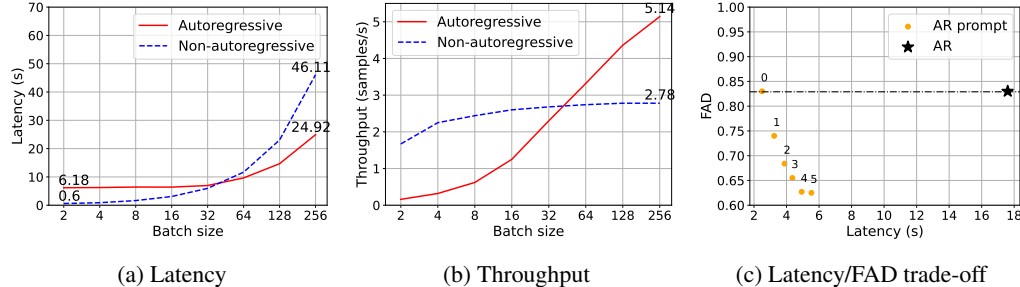

(a) Latency     (b) Throughput     (c) Latency/FAD trade-off

Figure 2: Latency and throughput analysis: MAGNET is particularly suited to small batch sizes (up to 10 times lower latency than MUSICGEN), while MUSICGEN benefits from a higher throughput for bigger batch sizes. MAGNET offers flexibility regarding the latency/quality trade off by allowing a customizable decoding schedule or following the Hybrid-MAGNET variant.

Table 1 presents the results of MAGNET on the task of text-to-music generation compared to various baselines. Results are reported on the MusicCaps benchmark. As can be seen, MAGNET reaches comparable performance to MusicGen, which performs autoregressive modeling, while being significantly faster both in terms of latency and decoding steps. When comparing to AudioLDM2, which is based on latent diffusion, MAGNET gets worse FAD and CLAP scores, while reaching better KL subjective scores. Notice, AudioLDM2 was trained on 10-seconds generation at 16kHz while MAGNET was trained on 30-seconds generation at 32kHz. When we reduce the sequence length to 10-second generations our FAD reaches to 2.9 and CLAP score of 0.31. We additionally evaluate MAGNET on the task of text-to-audio generation (environmental sound generation). Results and details regarding the baselines can be found in Appendix G. Results show similar trends as on text-to-music of MAGNET providing comparable performance to the autoregressive baseline while being significantly faster.

## 5.2 ANALYSIS

**Latency vs. Throughput.** We analyze the trade-offs between latency and throughput as a function of the batch size, as illustrated in Fig. 2a and Fig. 2b. Latency and throughput are reported for generated samples of 10-second duration on an A100 GPU with 40 GB of RAM. Due to CFG, the batch size value is typically twice the generated sample count. Indeed the model outputs two distributions in parallel: one conditioned on the text prompt and the other unconditioned.

Compared with the baseline autoregressive model (red curve), the non-autoregressive model (dashed blue curve) especially excels on small batch sizes due to parallel decoding, with a latency as low as 600 ms for a single generated sample (batch size of two in the 2a), more than 10 times faster than the autoregressive baseline. This is especially interesting in interactive applications that require low-latency. The non-autoregressive model is faster than the baseline up to a batch size of 64.

However, in scenarios where throughput is a priority (e.g. generate as many samples as possible, irrespective of the latency), we show that the autoregressive model is favorable. While the non-autoregressive model throughput is bounded to $\sim 2.8$ samples/second for batch sizes bigger than 64, the autoregressive model throughput is linear in batch size, only limited by the GPU memory.

**Hybrid-MAGNET.** Next, we demonstrate how a hybrid version can also be combined. We bootstrap the non-autoregressive generation with an autoregressive-generated audio prompt. We train a single model that incorporates both decoding strategies. During training, we sample a time step $t \in \{1, \ldots, T\}$ and compute the autoregressive training loss for all positions that precede $t$. The rest of the sequence is being optimized using the MAGNET objective. This is done by designing a custom attention mask that simulates the inference behavior during training (causal attention before $t$, parallel attention after $t$). During inference, the model can be used autoregressively to generate a short audio prompt and switch to non-autoregressive decoding to complete the generation faster. A detailed description of the hybrid training can be found on Appendix E.

We analyze the effect of the chosen $t$ in Fig. 2c using a 30-second generations without rescoring. Starting from a fully non-autoregressive generation decoding, we ablate on the autoregressive-generated prompt duration. The results indicate that the longer the prompt, the lower the FAD. The Hybrid-MAGNET is even able to outperform the full autoregressive baseline (when consid-

Table 2: Span length and restricted context ablation. We report FAD scores for MAGNET using an In-domain test set considering different span lengths, with and without temporally restricted context.

| Span-length | 1 | | 2 | | 3 | | 4 | | 5 | | 10 | |
|---|---|---|---|---|---|---|---|---|---|---|---|---|
| Restricted context | ✗ | ✓ | ✗ | ✓ | ✗ | ✓ | ✗ | ✓ | ✗ | ✓ | ✗ | ✓ |
| FAD | 3.07 | 2.13 | 0.74 | 0.66 | 0.82 | 0.61 | 0.97 | 0.63 | 1.13 | 0.84 | 1.66 | 1.05 |

Table 3: We evaluate the effect of the rescorer on model performance. We report mean and CI95.

| MODEL | RE-SCORE | $FAD_{vgg} \downarrow$ | $KL \downarrow$ | $CLAP_{scr} \uparrow$ | OVL. $\uparrow$ | REL. $\uparrow$ | LATENCY (S) |
|---|---|---|---|---|---|---|---|
| MAGNET-small | ✗ | 3.7 | 1.18 | 0.30 | $80.65{\pm}1.48$ | $81.06{\pm}1.19$ | 2.5 |
| MAGNET-small | ✓ | 3.3 | 1.12 | 0.31 | $83.31{\pm}1.11$ | $84.53{\pm}1.11$ | 4.0 |
| MAGNET-large | ✗ | 4.2 | 1.19 | 0.30 | $82.19{\pm}1.23$ | $82.96{\pm}0.91$ | 8.2 |
| MAGNET-large | ✓ | 4.0 | 1.15 | 0.29 | $84.43{\pm}1.24$ | $85.57{\pm}1.04$ | 12.6 |

ering FAD), starting from a 1-second prompt while still being significantly faster (3.2s of latency down from 17.6s). This Hybrid strategy is another way to control quality/latency trade-offs when performance of the non-autoregressive model does not quite match its autoregressive counterpart.

## 5.3 ABLATION

**The effect of modeling choices.** To validate our findings regarding the necessity of span masking for audio modeling, as well as the necessity of temporal context restriction for efficient optimization, we train different model configurations and report the resulting FAD in Table 2. Results suggest that using restricted context consistently improves model performance across all settings. Moreover, using a span-length of 3, which corresponds to spans of 60ms yields the best performance.

**The effect of CFG annealing.** Table 6 in the Appendix presents results computed over in-domain samples using several CFG coefficient configurations. We evaluate both constant CFG schedules, e.g. by setting $\lambda_0 = \lambda_1 = 3$, and annealing CFG. Results suggest that using $\lambda_0 = 10$, $\lambda_1 = 1$ yields the best FAD score over all evaluated setups. This finding aligns with our hypothesis that during the first decoding steps a stronger text adherence is required, while at later decoding steps we would like the model to focus on previously decoded tokens.

**The effect model rescorer.** Next, we evaluate the effect of model rescorering on the overall performance. Results are presented in Table 3. Results suggest that applying model rescoring improves performance for almost all metrics. However, this comes at the expense of slower inference time.

**The effect of decoding steps.** We explore the effect of less decoding steps on the overall latency and performance, see Fig. 7. It seems that reducing the decoding steps for higher levels does not impact quality as much as for the first level. For scenarios where minimizing the latency is the top priority, one should consider only 1 step per higher codebook level: in such case, latency drops to 370 ms, at the expense of a 8% increase of FAD compared to 10 steps per higher levels.

**Decoding visualization.** We visualize the masking dynamics of MAGNET's iterative decoding process. In specific, we plot the mask $m(i)$ chosen by MAGNET during the generation of a 10-second audio sample, for each iteration $i \in \{1, \ldots, 20\}$. As can be seen, MAGNET decodes the audio sequence in a non-causal manner, choosing first a sparse set of token spans at various disconnected temporal locations, and gradually "inpaint" the gaps until convergence to a full token sequence. Visualization and full details can be found in Appendix F.

## 6 RELATED WORK

**Autoregressive audio generation.** Recent studies considering text-to-audio generation can be roughly divided into two: (i) environmental sounds generation; and (ii) music generation. As for environmental sound generation, Kreuk et al. (2022a) proposed applying a transformer language model over discrete audio representation, obtained by quantizing directly time-domain signals using EnCodec Défossez et al. (2022). Sheffer & Adi (2023) followed a similar approach to Kreuk et al. (2022a) for image-to-audio generation. Dhariwal et al. (2020) proposed representing music samples

in multiple streams of discrete representations using a hierarchical VQ-VAE. Next, two sparse transformers applied over the sequences to generate music. Gan et al. (2020) proposed generating music for a given video, while predicting its midi notes. Recently, Agostinelli et al. (2023) proposed a similar approach to AudioLM (Borsos et al., 2023a), which represents music using multiple streams of "semantic tokens" and "acoustic tokens". Then, they applied a cascade of transformer decoders conditioned on a textual-music joint representation (Huang et al., 2022). Donahue et al. (2023) followed a similar modeling approach, but for the task of singing-to-accompaniment generation. Copet et al. (2023) proposed a single stage Transformer-based autoregressive model for music generation, conditioned on either text or melodic features, based on EnCodec .

**Non-autoregressive audio generation.** The most common approach for non-autoregressive generation is diffusion models. These models naturally apply over continuous representations however can also operate over discrete representations. Yang et al. (2022) proposed representing audio spectrograms using a VQ-VAE, then applying a discrete diffusion model conditioned on textual CLIP embeddings for the generation part Radford et al. (2021). Huang et al. (2023b); Liu et al. (2023a;b) proposed using latent diffusion models for the task of text-to-audio, while extending it to various other tasks such as inpainting, image-to-audio, etc. Schneider et al. (2023); Huang et al. (2023a); Maina (2023); Forsgren & Martiros (2022); Liu et al. (2023b) proposed using a latent diffusion model for the task of text-to-music. Schneider et al. (2023) proposed using diffusion models for both audio encoder-decoder and latent generation. Huang et al. (2023a) proposed a cascade of diffusion model to generate audio and gradually increase its sampling rate. Forsgren & Martiros (2022) proposed fine-tuning Stable Diffusion (Rombach et al., 2022) using spectrograms to generate five-second segments, then, using image-to-image mapping and latent interpolation to generate long sequences. Li et al. (2023) present impressive generation results using a latent diffusion model with a multi-task training objective, however for 10-second generation only.

The most relevant prior work to ours involves masked generative modeling. Ghazvininejad et al. (2019) first proposed the Mask-Predict method, a masked language modeling with parallel decoding for the task of machine translation. Later on, Chang et al. (2022) followed a similar modeling strategy, denoted as MaskGIT, for the task of class-conditioned image synthesis and image editing, while Chang et al. (2023) extended this approach to high-quality textually guided image generation over low-resolution images followed by a super-resolution module. Lezama et al. (2022) further proposed the TokenCritic approach, which improves the sampling from the joint distribution of visual tokens over MaskGIT. Recently, Borsos et al. (2023b) proposed the SoundStorm model, which has a similar modeling strategy as MaskGIT but for text-to-speech and dialogue synthesis. Unlike MaskGIT, the SoundStorm model is conditioned on semantic tokens obtained from an autoregressive model. The proposed work differs from this model as we propose a single non-autoregressive model, with a novel audio-tokens modeling approach for the task of text-to-audio. Another concurrent work, is VampNet (Garcia et al., 2023), a non-autoregressive music generation model. Unlike MAGNET, VampNet is based on two different models (one to model the "coarse" tokens and one to model the "fine" tokens), and do not explore text-to-music generation without audio-prompting.

## 7 DISCUSSION

**Limitations.** As discussed in section 5.2, the proposed non-autoregressive architecture targets low-latency scenarios. By design, the model re-encodes the whole sequence at each decoding step, even for time steps that have not changed between two consecutive decoding steps. This is a fundamental difference with autoregressive architectures that can benefit from caching past keys and values and only encode one-time step per decoding step, which efficiently scales when the batch size increases. Such a caching strategy could also be adopted for non-autoregressive architectures, for time steps that do not change between consecutive decoding steps, however, this requires further research.

**Conclusion.** In this work, we presented MAGNET which, to the best of our knowledge, is the first pure non-autoregressive method for text-conditioned audio generation. By using a single-stage encoder during training and a rescorer model, we achieve competitive performance with autoregressive methods while being approximately 7 times faster. We also explore a hybrid approach that combines autoregressive and non-autoregressive models. Our extensive evaluation, including objective metrics and human studies, highlights MAGNET's promise for real-time audio generation with comparable or minor quality degradation. For future work, we intend to extend the research work on the model rescoring and advanced inference methods. We believe this research direction holds great potential in incorporating external scoring models which will allow better non-left-to-right model decoding.

ACKNOWLEDGEMENTS.

The authors would like to thank Or Tal, Michael Hassid and Nitay Arcusin for the useful theoretical discussions. The authors would additionally like to thank Kamila Benzina and Nisha Deo for supporting this project. This research work was supported in part by ISF grant 2049/22.

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

Table 4: Details about the training sets used to train the proposed method and the evaluated baselines.

| METHOD | NO. OF HOURS | SAMPLING RATE | DETAILS |
|---|---|---|---|
| MusicGen | 20,000 | 32kHz | ShutterStock, Pond5, and proprietary data |
| MusicLM | 280,000 | 24 kHz | Proprietary data |
| Mousai | 2,500 | 48kHz | ShutterStock, Pond5, and proprietary data |
| 'AudioLDM2 | 29,510 | 16kHz | AudioSet, WavCaps, AudioCaps, VGGSound, Free Music Archive, Million Song Dataset, LJSpeech, and GigaSpeech |
| MAGNET | 20,000 | 32kHz | ShutterStock, Pond5, and proprietary data |

# A  EXPERIMENTAL SETUP

## A.1  IMPLEMENTATION DETAILS

Under all setups, we use the official EnCodec model as was published by Copet et al. (2023)[7]. The model gets as input an audio segment and outputs a 50 Hz discrete representation. We use four codebooks where each has a codebook size of $2048$. We perform the same text preprocessing as proposed by Copet et al. (2023); Kreuk et al. (2022a).

We train non-autoregressive transformer models using 300M (MAGNET-small) and 1.5B (MAGNET-large) parameters. We use a memory efficient Flash attention (Dao et al., 2022) from the xFormers package (Lefaudeux et al., 2022) to improve both speed and memory usage. We train models using 30-second audio crops sampled at random from the full track. We train the models for 1M steps with the AdamW optimizer (Loshchilov & Hutter, 2017), a batch size of 192 examples, $\beta_1 = 0.9$, $\beta_2 = 0.95$, a decoupled weight decay of 0.1 and gradient clipping of 1.0. We further rely on D-Adaptation based automatic step-sizes (Defazio & Mishchenko, 2023). We use a cosine learning rate schedule with a warmup of 4K steps. Additionally, we use an exponential moving average with a decay of 0.99. We train the models using respectively 32 GPUs for the small model and 64 GPUs for the large ones with float16 precision.

Finally, for inference, we employ nucleus sampling (Holtzman et al., 2020) with top-p 0.9, and a temperature of 3.0 that is linearly annealed to zero during decoding iterations. We use CFG with condition dropout rate of 0.3 during training, and a guidance coefficient 10.0 that is annealed to 1.0 during iterative decoding.

## A.2  DATASETS

We follow the same setup as in Copet et al. (2023) and use 20K hours of licensed music to train MAGNET. Specifically, we rely on the same 10K high-quality music tracks, the ShutterStock, and Pond5 music data collections as used in Copet et al. (2023)[8] with respectively 25K and 365K instrument-only music tracks. All datasets consist of full-length music sampled at 32 kHz with metadata composed of a textual description and additional information such as the genre, BPM, and tags.

For the main results and comparison with prior work, we evaluate the proposed method on the MusicCaps benchmark (Agostinelli et al., 2023). MusicCaps is composed of 5.5K samples (ten-second long) prepared by expert musicians and a 1K subset balanced across genres. We report objective metrics on the unbalanced set, while we sample examples from the genre-balanced set for qualitative evaluations. We additionally evaluate the proposed method using the same in-domain test set as proposed by Copet et al. (2023). All ablation studies were conducted on the in-domain test set.

---

[7]https://github.com/facebookresearch/audiocraft
[8]www.shutterstock.com/music and www.pond5.com

### A.3 EVALUATION

**Baselines.** For music generation we compare MAGNET Mousai (Schneider et al., 2023), Music-Gen Copet et al. (2023), AudioLDM2 Liu et al. (2023b), and MusicLM (Agostinelli et al., 2023). For Mousai, we train a model using our dataset for a fair comparison, using the open source implementation provided by the authors[9].

**Evaluation metrics.** We evaluate the proposed method using the same setup as proposed in Copet et al. (2023); Kreuk et al. (2022b), which consists of both objective and subjective metrics. For the objective methods, we use three metrics: the Fréchet Audio Distance (FAD), the Kullback-Leiber Divergence (KL) and the CLAP score (CLAP). We report the FAD (Kilgour et al., 2018) using the official implementation in Tensorflow with the VGGish model [10]. A low FAD score indicates the generated audio is plausible. Following Kreuk et al. (2022a), we use a state-of-the-art audio classifier trained for classification on AudioSet (Koutini et al., 2021) to compute the KL-divergence over the probabilities of the labels between the original and the generated audio. For the music generation experiments only we additionally report the CLAP score (Wu et al., 2023; Huang et al., 2023b) between the track description and the generated audio to quantify audio-text alignment, using the official pretrained CLAP model [11].

For the human studies, we follow the same setup as in Kreuk et al. (2022a). We ask human raters to evaluate two aspects of the audio samples (i) overall quality (OVL), and (ii) relevance to the text input (REL). For the overall quality test, raters were asked to rate the perceptual quality of the provided samples in a range of 1 to 100. For the text relevance test, raters were asked to rate the match between audio and text on a scale of 1 to 100. Raters were recruited using the Amazon Mechanical Turk platform. We evaluate randomly sampled files, where each sample was evaluated by at least 5 raters. We use the CrowdMOS package[12] to filter noisy annotations and outliers. We remove annotators who did not listen to the full recordings, annotators who rate the reference recordings less than 85, and the rest of the recommended recipes from CrowdMOS (Ribeiro et al., 2011). For fairness, we include the same normalization scheme as proposed in Copet et al. (2023) of normalizing all samples at $-14$dB LUFS.

## B RECEPTIVE FIELD ANALYSIS

We present the receptive field analysis of the EnCodec model in Fig. 3. We slide an impulse function, in the form of a one-hot input vector, and measure the norm of the encoded latent vector in the middle of the sequence, as function of the temporal distance from the impulse. We perform the process twice: (i) For the full encoder (left) and (ii) while omitting the LSTM block and remaining only with the convolutional network (right). Fig. 3 shows that the effective receptive field of EnCodec is upper bounded by 100ms in each direction, supporting our choice to design MAGNET's restricted transformer s.t. codebooks greater than one attend only tokens in a neighborhood of 100ms in each direction.

## C SPAN MASKING

Sampling a placement of $u$ token spans can be implemented by first sampling a subset of $u$ indices from $\{1, \ldots, T\}$, serving as the span starts, and then extending each index to a span. Formally, we sample $I^{(u)} \sim \mathcal{U}(\{\mathcal{A} \subseteq \{1, \ldots, T\} : |\mathcal{A}| = u\})$, and then extend each index $t \in I^{(u)}$ to the span of indices $t, \ldots, t + l - 1$. The total set of masked indices would be

$$\mathcal{M}^{\text{spans}}(I^{(u)}; l) \triangleq \bigcup_{t \in I^{(u)}} \{t, \ldots, t + l - 1\}. \tag{7}$$

---

[9]Implementation from github.com/archinetai/audio-diffusion-pytorch (March 2023)

[10]github.com/google-research/google-research/tree/master/frechet_audio_distance

[11]https://github.com/LAION-AI/CLAP

[12]http://www.crowdmos.org/download/

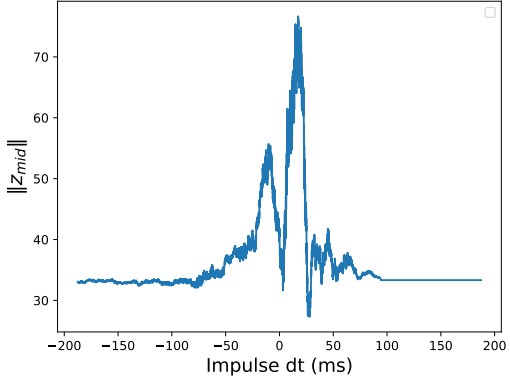
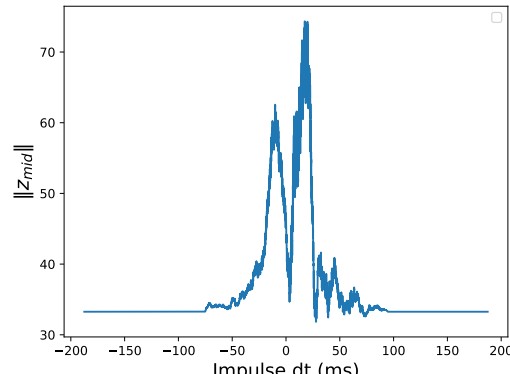

(a) EnCodec's middle latent vector's impulse response.

(b) The impulse response of the same vector when omitting the LSTM block from the encoder.

Figure 3: A visualization of the receptive field analysis.

**Proposition C.1.** *Given a random placement of $u$ spans of size $l$ over a sequence of length $T$, the expected masking rate is as follows,*

$$\mathbb{E}_{I^{(u)} \sim \mathcal{U}(\{\mathcal{A} \subseteq \{1, \ldots, T\} : |\mathcal{A}| = u\})} \left[ \frac{1}{T} \left| \mathcal{M}^{spans} \left( I^{(u)}; l \right) \right| \right] = 1 - \frac{\binom{T-l}{u}}{\binom{T}{u}}. \tag{8}$$

**Derivation:** First, note that for a given token $z_t$, the probability that $z_t$ would remain unmasked, is the probability to choose $u$ span starts only from the indices:

$$A_t \triangleq \{1, \ldots, T\} \setminus \{t - l + 1, \ldots, t\}. \tag{9}$$

The total number of placements is $\binom{T}{u}$, i.e., the number of possibilities to choose $u$ span starts out of a set of $T$ indices without replacement. Similarly, the total amount of placements for which all span starts are in $A_t$, is $\binom{T-l}{u}$. Thus,

$$\mathbb{P}\left[ t \in \mathcal{M}^{spans}(I^{(u)}; l) \right] = 1 - \frac{\binom{T-l}{u}}{\binom{T}{u}}. \tag{10}$$

Consequently, the masking probability for each token is $1 - \binom{T-l}{u}/\binom{T}{u}$. Finally, we define the indicator random variable $\mathbb{1}_{t \in \mathcal{M}^{spans}(I^{(u)};l)}$ for each $t \in \{1 \ldots T\}$, and conclude the derivation by

$$\mathbb{E}_{I^{(u)}} \left[ \left| \mathcal{M}^{spans} \left( I^{(u)}; l \right) \right| \right] = \mathbb{E}_{I^{(u)}} \left[ \sum_{t=1}^{T} \mathbb{1}_{t \in \mathcal{M}^{spans}(I^{(u)};l)} \right] \tag{11}$$

$$= \sum_{t=1}^{T} \mathbb{E}_{I^{(u)}} \left[ \mathbb{1}_{t \in \mathcal{M}^{spans}(I^{(u)};l)} \right] \tag{12}$$

$$= T \cdot \left( 1 - \frac{\binom{T-l}{u}}{\binom{T}{u}} \right). \tag{13}$$

## D    MODEL INFERENCE

Fig. 4 presents the inference process of MAGNET. For clarity, we omit CFG and nucleus sampling, and assume $T$ is a multiple of the span length $l$. To further ease the reading, we present the inference algorithm for a single codebook, while in practice, we run Fig. 4 for every codebook $k \in \{1 \ldots K\}$.

```python
def MAGNeT_generate(B: int, T: int, text: List, s: int, model: nn.Module,
                    rescorer: nn.Module, mask_id: int, tempr: float, w: float):

    # Start from a fully masked sequence
    gen_seq = torch.full((B, T), mask_id, dtype=torch.long)

    n_spans = T // span_len
    spans_shape = (B, n_spans)
    span_scores = torch.zeros(spans_shape, dtype=torch.float32)

    # Run MAGNeT iterative decoding for 's' iterations
    for i in range(s):
        mask_p = torch.cos((math.pi * i) / (2 * s))
        n_masked_spans = max(int(mask_p * n_spans), 1)

        # Masking
        masked_spans = span_scores.topk(n_masked_spans, dim=-1).indices
        mask = get_mask(spans_shape, masked_spans)
        gen_seq[mask] = mask_id

        # Forward pass
        logits, probs = model.compute_predictions(gen_sequence, text, cfg=True, temperature=tempr)

        # Classifier free guidance with annealing
        cfg_logits = cfg(mask_p, logits, annealing=True)

        # Sampling
        sampled_tokens = sample_top_p(probs, p=top_p)

        # Place the sampled tokens in the masked positions
        mask = gen_seq == mask_id
        gen_seq = place_sampled_tokens(mask, sampled_tokens[..., 0], gen_seq)

        # Probs of sampled tokens
        sampled_probs = get_sampled_probs(probs, sampled_tokens)

        if rescorer:
            # Rescoring
            rescorer_logits, rescorer_probs = rescorer.compute_predictions(gen_seq, text)
            rescorer_sampled_probs = get_sampled_probs(rescorer_probs, sampled_tokens)

            # Final probs are the convex combination of probs and rescorer_probs
            sampled_probs = w * rescorer_sampled_probs + (1 - w) * sampled_probs

        # Span scoring - max
        span_scores = get_spans_scores(sampled_probs)

        # Prevent remasking by placing -inf scores for unmasked
        span_scores = span_scores.masked_fill(~spans_mask, -1e5)

    return gen_seq
```

Figure 4: MAGNET's text-to-audio inference. MAGNET performs iterative decoding of $s$ steps. In each step, the least probable non-overlapping spans are being masked, where the probability is a convex combination of the restricted-transformer confidence and the probability obtained by the pre-trained rescorer. Finally, the span probabilities are re-updated, while assigning $\infty$ to the unmasked spans, to prevent its re-masking and fix it as anchors for future iterations.

## E   HYBRID-MAGNET TRAINING

The aim of Hybrid-MAGNET is to switch from autoregressive generation to non-autoregressive during inference, so as to generate an audio prompt with the same quality as MUSICGEN that can be completed fast using MAGNET inference. The goal is to find a compromise between MUSICGEN quality and MAGNET speed. To give Hybrid-MAGNET the ability to switch between decoding strategies, it requires a few adaptations from MAGNET training recipe. One of them is to train jointly on two different objectives as illustrated by Figure 5. Similarly to Borsos et al. (2023b) a time step $t$ is uniformly sampled from $\{1, \ldots, T\}$ that simulates an audio prompt for MAGNET to complete from. For all positions that precede $t$ and for all codebook levels we propose to compute the autoregressive training objective, using causal attention masking. For all succeeding positions we keep the MAGNET training objective: the model can attend to tokens from the audio prompt. Moreover the codebook pattern is adapted for the autoregressive generation to work well, to that

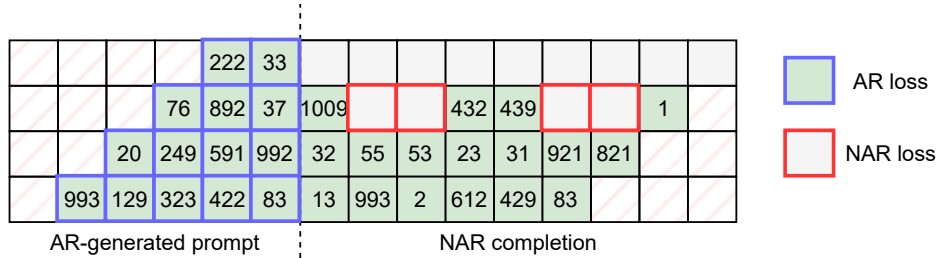

Figure 5: Training of Hybrid-MAGNET. During training a random timestep $t$ is sampled. For timesteps preceding $t$ a causal attention mask is applied and cross entropy loss is computed for all levels (blue highlighted squares). For timesteps succeeding $t$ the standard MAGNET training strategy is applied. Codebook levels are shifted following the *delay* pattern from Copet et al. (2023).

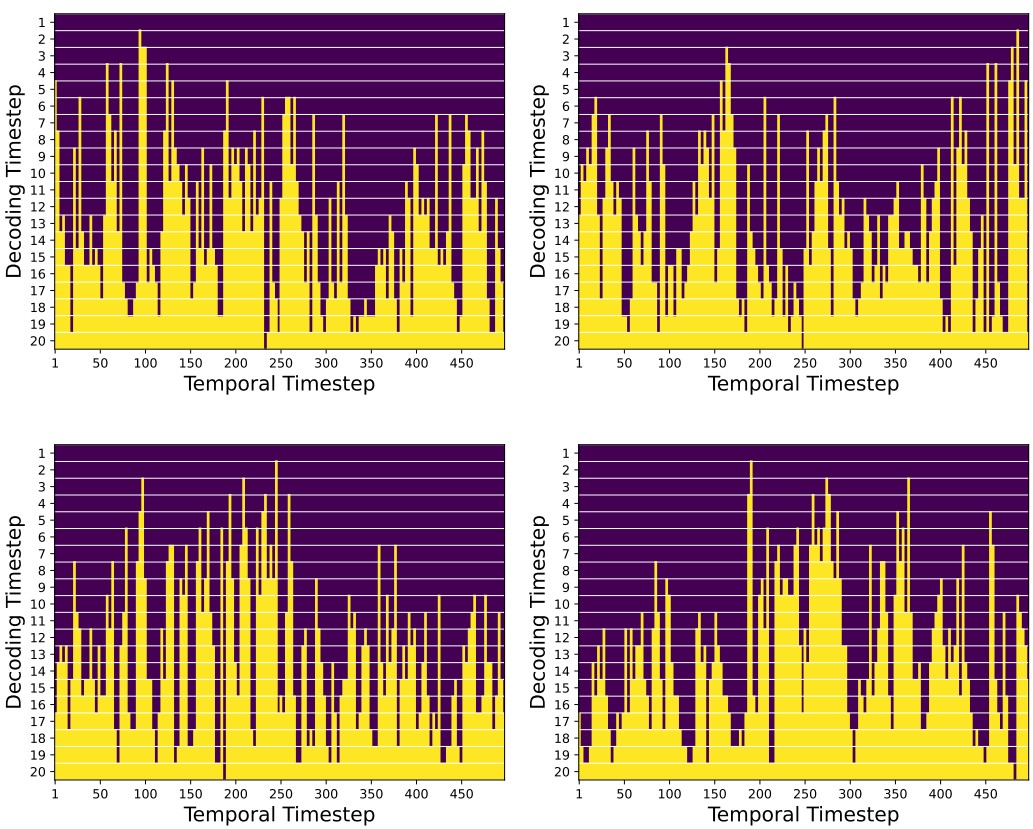

Figure 6: Decoding visualization of the chosen anchor tokens as a function of decoding steps, for an iterative decoding process with $s = 20$. We plot the mask $m(i)$ chosen by MAGNET, for each $i \in \{1, \ldots, s\}$, during the generation of a 10-second audio sample for the text prompt 'A dynamic blend of hip-hop and orchestral elements, with sweeping strings and brass'. The x-axis represents time while the y-axis represents the decoding steps.

end we use the *delay* pattern from Copet et al. (2023). Thus the temporally restricted context from MAGNET is adapted to take into account the codebook level-dependent shifts.

Table 5: Text-to-Audio generation results. We report FAD and KL scores for all methods.

| | PARAMETERS | TEXT CONDITIONING | FAD↓ | KL↓ |
|---|---|---|---|---|
| DiffSound | 400M | CLIP | 7.39 | 2.57 |
| AudioGen-base | 285M | T5-base | 3.13 | 2.09 |
| AudioGen-large | 1500M | T5-large | 1.77 | 1.58 |
| AudioLDM2-small | 346M | T5-large, CLAP, ImageBind, PE | 1.67 | 1.01 |
| AudioLDM2-large | 712M | T5-large, CLAP, ImageBind, PE | 1.42 | 0.98 |
| Make-an-Audio | 332M | CLAP | 4.61 | 2.79 |
| MAGNET-small | 300M | T5-large | 3.22 | 1.42 |
| MAGNET-large | 1500M | T5-large | 2.36 | 1.64 |

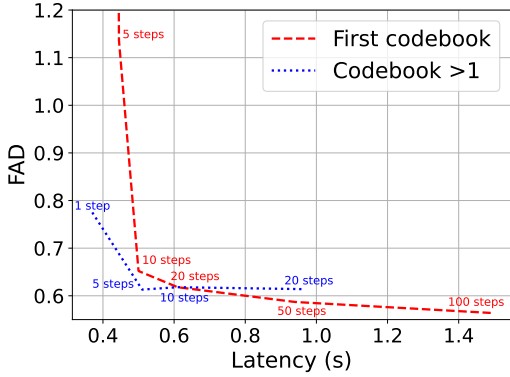

Figure 7: Effect of the decoding schedules on the quality/latency trade off. We vary the number of decoding steps for the first codebook level (dashed red curve) and the higher codebook levels (dotted blue curve) around a $(20, 10, 10, 10)$ decoding schedule.

## F   ITERATIVE DECODING DYNAMICS

Fig. 6 presents the masking dynamics of MAGNET's iterative decoding process with $s = 20$. In specific, we plot the mask $m(i)$ chosen by MAGNET, for each $i \in \{1 \ldots s\}$, during the generation of a 10-second audio sample for the text prompt *'A dynamic blend of hip-hop and orchestral elements, with sweeping strings and brass'*. To demonstrate MAGNET's stochasticity, we repeat the process several times. As can be seen, MAGNET decodes the audio sequence in a non-causal manner, choosing first a sparse set of token-spans at various disconnected temporal locations, and gradually "inpaint" the gaps until convergence to a full token sequence.

## G   ADDITIONAL RESULTS

**Text-to-audio generation** We follow Kreuk et al. (2022a) and use the exact same training sets to optimize MAGNET. We train MAGNET in two model scales, of 300M and 1.5B parameters respectively, and compare it to AudioGen (Kreuk et al., 2022a), DiffSound (Yang et al., 2022), AudioLDM2 (Liu et al., 2023b), and Make-an-Audio Huang et al. (2023b). Results are reported in Table 5. Results are reported on the AudioCaps testset (Kim et al., 2019). All audio files were sampled at 16kHz. As can be see MAGNET results are comparable or slightly worse than the autoregressive alternative (AudioGen) while having significantly lower latency (the latency values are the same as in Table 1 for MAGNeT, while AudioGen has the same latency as MusicGen). For inference, different than the MAGNET models trained for music generation, we use top-p 0.8, an initial temperature of 3.5, and an initial CFG guidance coefficient of 20.0.

**The effect of decoding steps.** The latency of the non-autoregressive model can be controlled by configuring the appropriate decoding steps, at the expense of quality degradation. In Fig. 7, we report the in-domain FAD as a function of latency for different decoding steps. We ablate on the first codebook level step count (dashed red curve) and the higher codebook levels step count (dotted

Figure 8: We restrict the attention maps to focus on local context for codebooks levels greater than 1. In this figure we consider 2 time-steps restrictions for each side, in practice we use 5 time-steps for each side, resulting in 11 tokens.

Table 6: CFG annealing ablation. We report FAD scores for different $\lambda_0, \lambda_1$ configurations, as well as KL and CLAP scores.

| $\lambda_0 \to \lambda_1$ | $1 \to 1$ | $3 \to 3$ | $10 \to 10$ | $10 \to 1$ | $20 \to 20$ | $20 \to 1$ |
|---|---|---|---|---|---|---|
| $\text{FAD}_{\text{vgg}} \downarrow$ | 3.95 | 0.99 | 0.63 | 0.61 | 0.80 | 0.68 |
| $\text{KL} \downarrow$ | 0.79 | 0.60 | 0.56 | 0.56 | 0.57 | 0.56 |
| $\text{CLAP}_{\text{scr}} \uparrow$ | 0.13 | 0.28 | 0.28 | 0.31 | 0.31 | 0.31 |

blue curve), starting from the $(20, 10, 10, 10)$ decoding schedule. The red curve illustrates that a good compromise between latency and quality can be obtained around 20 steps for the first level, after which decoding further will only marginally lower the FAD, while adding latency. Regarding the higher codebook levels, we can observe an inflection point happening around 5 steps after which FAD remains stable. It is interesting to note that reducing the decoding steps for higher levels does not impact quality as much as for the first level. For example the $(10, 10, 10, 10)$ decoding schedule achieves $0.65$ FAD at a latency close to that of the $(20, 5, 5, 5)$ schedule, which achieves a lower FAD of $0.61$ despite a smaller total step count.

**The effect of CFG guidance annealing.** We evaluate both constant CFG schedules, e.g. by setting $\lambda_0 = \lambda_1 = 3$, and annealing CFG. Results are presented in Table 6. Results suggest that using $\lambda_0 = 10, \lambda_1 = 1$ yields the best FAD score over all evaluated setups. This finding aligns with our hypothesis that during the first decoding steps a stronger text adherence is required, while at later decoding steps we would like the model to put more focus on previously decoded tokens.

