# OpenReview forum: "Masked Audio Generation using a Single Non-Autoregressive Transformer"
_ICLR.cc/2024/Conference — ICLR 2024 poster_

### Official Review · Reviewer_f8w3 · 2023-10-29

**Soundness:** 3 good
**Presentation:** 3 good
**Contribution:** 3 good
**Rating:** 8
**Confidence:** 4

**Summary:**

The proposed model (MAGNET) is a generative sequence modeling method for discrete audio tokens. It uses a single-stage, non-autoregressive transformer encoder during training to predict masked token spans and gradually constructs the output sequence during inference. The authors further introduce a model rescorer that leverages another pretrained model to rank predictions. A hybrid version combines autoregressive and non-autoregressive models, generating the initial part autoregressively and the rest in parallel. MAGNET operates on a multi-stream audio representation (EnCodec) and uses a single transformer model trained to predict spans of masked input tokens. During inference, it constructs the output audio sequence through multiple decoding steps. The method achieves competitive results in text-to-music and text-to-audio generation, with seven times faster latency compared to autoregressive methods, and offers insights through ablation studies.

**Strengths:**

The authors of this research paper address the challenging problem of generating long audio and music sequences, a task that has garnered significant attention due to its relevance in various applications, such as text-to-music and text-to-audio generation. Their proposed solution, known as MAGNET (Masked Audio Generation using Non-autoregressive Transformer), introduces several innovative techniques to tackle this problem effectively.

One key feature of MAGNET is its use of a training via masking strategy, a popular approach in contemporary self-supervised learning. This approach is particularly well-suited for the task of audio generation as it helps the model to learn dependencies and patterns in the data by predicting masked segments. In the case of MAGNET, the authors employ span masking, a technique rooted in speech and audio processing, which proves to be more effective than traditional token based masking. Span masking mitigates the issue of information leakage and results in more coherent and high-quality audio generation.

MAGNET's underlying representation is based on EnCodec, which combines full self-attention with smaller attention spans derived from the impulse response characteristics of LSTM blocks. This hybrid approach allows the model to efficiently capture long-range dependencies while preserving local details, contributing to the generation of realistic audio sequences.

A notable innovation in this work is the introduction of model rescoring through MusicGen and AudioGen. This technique enhances the robustness of the model during the sampling process, ensuring that the generated audio maintains high quality and coherence. Additionally, the authors incorporate annealing in the classifier-free guidance parameter, a strategy previously demonstrated to be useful in 3-D shape generation. This further contributes to the model's performance.

The authors conduct a comprehensive evaluation of MAGNET on both text-to-music and text-to-audio generation tasks. They use objective metrics such as FAD, KL Divergence, and CLAP scores to quantitatively assess the model's performance. Subjective metrics, derived from crowd-sourced evaluations and including Mean Opinion Score (MOS) and audio-text relevance assessments, provide a more human-centered perspective on the quality of the generated audio.

To further validate their approach and elucidate the rationale behind their design choices, the authors conduct ablation studies. These studies help in understanding the individual contributions of different components to the overall performance of MAGNET.

Finally, the authors perform a comparison of MAGNET with existing methods, including MusicGen, Mousai, and AudioLDM2, providing a clear understanding of its strengths and weaknesses in comparison to state-of-the-art approaches.

**Weaknesses:**

The main weakness of this paper is that there is a heavy amount of engineering involved in the development of this model. While, it is definitely commendable, it makes reproducing the result extremely difficult for the rest of research community. Further, authors have not referred to this work titled "Masked Autoencoders that Listen" which has a very similar idea of span masking. It  would have been interesting to see the contrast and comparison against something that is designed with similar idea. Another issue I have with the paper is that the authors do not provide any details about the dataset used for training.

**Questions:**

NA

---

> ### Author Response · Authors · 2023-11-17
> **Official author's response**
>
> **Regarding reproducibility of the proposed approach:** We believe all the required details to reproduce the results of the proposed approach can be found in the manuscript. Nevertheless, we will open-source all the source code and pre-trained models for full reproducibility by the research community.
>
> **Regarding reference to AudioMAE:** Thanks for pointing out the Masked Auto-Encoder that Listen paper [1]. We agree that both methods are conducting masking using spans of input representations, however, the motivation behind these two methods are different: (i) AudioMAE is based on masking patches of spectrogram representation of the input signal, while the proposed approach operates on discrete representations obtained from EnCodec; (ii) The goal of AudioMAE is to learn a semantic representation which will useful for downstream classification tasks such as audio event classification, speech classification tasks, etc. Our goal is to optimize a generative model that will be used to generate music/audio given textual description. We included a reference to AudioMAE with a short discussion in the related work section.
>
> **Regarding the used datasets:** We provide details about the dataset used in this work, their size, etc. please see Appendix A.2, the datasets section.
>
> [1] Huang, Po-Yao, et al. "Masked autoencoders that listen." Advances in Neural Information Processing Systems 35 (2022): 28708-28720.

---

### Official Review · Reviewer_4Cc9 · 2023-10-31

**Soundness:** 4 excellent
**Presentation:** 3 good
**Contribution:** 4 excellent
**Rating:** 8
**Confidence:** 5

**Summary:**

MAGNeT constitutes a mask generative modeling approach for conditional audio synthesis, leveraging the neural audio codec (EnCodec) to transform audio waveforms into compressed and discrete sequences. The main contribution lies in transfering the masked generative modeling, as exemplified by MaskGIT in the image domain, to the audio domain. In my opinion, the fact that the methods are shown to be effective in other domains (image, for example) does not necessarily undermine the findings in this work. My assessment therefore is based on completeness and rigor in justification of design choices and experimental details.

**Strengths:**

The choice of spanned mask prediction training with restricted context for RVQ is well justified both intuitively and through ablation studies. The use of a shifted impulse function for analyzing EnCodec's latent vector is a welcomed addition, especially to better understand the dependency between multi-level RVQ tokens, which will inspire the community to improve the methods to train the codec model and/or the generative model alike.

The accuracy-latency tradeoff analysis for the first-level RVQ token versus the rest is also helpful in designing a better decoding strategy for the RVQ tokens. It is worth mentioning that such dynamics can potentially change when an improved codec model is developed. The analysis methods in MAGNet are generalizable to assess future-generation codec models as well, and will likely become useful tools to design an optimal generative model for the target codec model.

**Weaknesses:**

The ablation study is satisfactory overall, but I would also like to see quantitative analysis on the proposed annealed classifier-free guidance scale, which seems to be missing in the current manuscript. While intuitive, documenting the performance gain obtained by the method would make the paper more convincing and complete.

Although the manuscript states that the training dataset is the same for several baseline models (Mousai and MusicGen), others (MusicLM and AudioLDM2) are trained on different data. Understanding the challenge in matching the music dataset (and open sourcing the data due to copyright concerns), further clarification on the training dataset for all baseline models via a complete paragraph and/or the table will be beneficial to gain a complete picture in the field of audio/music generative models.

**Questions:**

There have been several concurrent works using a similar approach, including SoundStorm, which is adequately mentioned in the manuscript. I would like to mention another concurrent work, VampNet[1], which applied MaskGIT-like training and a neural audio codec for music generation. While the comparison is not strictly necessary, it would be beneficial for the audience to have a summary of the differences between the models, such as highlighting the spanned mask prediction training and rescoring method of MAGNeT.

[1] Garcia, Hugo Flores, et al. "Vampnet: Music generation via masked acoustic token modeling." arXiv preprint arXiv:2307.04686 (2023).

---

> ### Author Response · Authors · 2023-11-17
> **Official author's response (part 1)**
>
> **Regarding CFG annealing ablation:** In the Table below we provide results of the proposed method considering different settings of CFG. We report FAD, KL, and CLAPScores. Results suggest that while some settings provide comparable performance considering one metric, following CFG annealing using 10->1 provides the best results considering all metrics.
>
> $\lambda_0 \rightarrow \lambda_1$      | 1 $\rightarrow$ 1 | 3 $\rightarrow$ 3 | 10 $\rightarrow$ 10 | 10 $\rightarrow$ 1 | 20 $\rightarrow$ 20 | 20 $\rightarrow$ 1 |
> |----------------------------------------|-------------------|-------------------|---------------------|--------------------|---------------------|--------------------|
> | Fad$_{\text{vgg}} \downarrow$ |  3.95  | 0.99  | 0.63  | **0.61**   | 0.80   | 0.68 |
> | Kl $\downarrow$               | 0.79 | 0.60 | 0.56   | 0.56  | 0.57   | 0.56 |
> | Clap$_{\text{scr}} \uparrow$   | 0.13 | 0.28 | 0.28   | 0.31  | 0.31   | 0.31 |
>
>
> **Regarding datasets details and comparison:** Thanks for the suggestion! Please see the Table below, we included this table in the appendix.
>
>
> | Method    | no. of Hours | Sampling rate | Source                                                                                               |
> |-----------|--------------|---------------|------------------------------------------------------------------------------------------------------------|
> | MusicGen  | 20,000       | 32kHz         | ShutterStock, Pond5, proprietary data                                                                      |
> | MusicLM   | 280,000      | 24 kHz        | Proprietary data                                                                                                    |
> | Mousai      | 20,000        | 32kHz         | ShutterStock, Pond5, proprietary data                                                                                                                                                                         |
> | AudioLDM2 | 29,510       | 16kHz         | AudioSet, WavCaps, AudioCaps, VGGSound, Free Music Archive, Million Song Dataset, LJSpeech, and GigaSpeech |
> | MAGNeT    | 20,000       | 32kHz          | ShutterStock, Pond5, proprietary data                                                                      |

---

> ### Author Response · Authors · 2023-11-17
> **Official author's response (part 2)**
>
> **Regarding including additional prior work:** Thanks for pointing out this concurrent work.  We would like to emphasize the main differences between SoundStorm and VampNet to the proposed method.
>
> In the VampNet model, the authors presented results for music inpainting and super-resolution and did not explore full-blown text-to-music generation without an additional **audio-prompting**. Moreover, unlike the proposed method VampNet is based on two different models, one for the "coarse” tokens and one for the “fine” tokens. MAGNeT uses a single model to generate all tokens. We updated the manuscript to include VampNet in the related work.
>
> As of SoundStrom, it was proposed to speed up inference time for the AudioLM [1] model, while the authors presented results for the task of text-to-speech only. As it is based on AudioLM, SoundStorm first models semantic tokens in an AR manner, specifically using w2v-bert tokens [2]. Hence, it is not clear how to convert such a model to text-to-music generation considering the semantic tokens needed for the task. Nevertheless, following the reviewer's request, we trained a version of SoundStorm, using the exact setup as described in the original soundstorm paper while we consider the first EnCodec codebook as semantic tokens. Results are reported below. As can be seen MAGNeT performs better across all metrics. Notice, the number of steps of the SoundStorm model is significantly higher than MAGNeT due to its autoregressive modeling of the first codebook.
>
>
> |Model  | Fad$_{vgg} \downarrow$ | Kl $\downarrow$ | Clap$_{scr} \uparrow$ | # Steps | Latency (s) |
> |-----------------|----------------------------------------|--------------------------|---------------------------------------|--------------------------|--------------------------|
> | Mousai          | 7.5                                    | 1.59               | 0.23                            | 200               | 44.0                 |
> | MusicLM         | 4.0                                    | -                        | -                                     | -                 | -                    |
> | AudioLDM 2      | 3.1                                    | 1.20               | 0.31                                    | 208               | 18.1                 |
> | MusicGen-small | 3.1                                    | 1.29               | 0.31                              | 1500              | 17.6                 |
> | Musicgen-large | 3.4                                    | 1.23               | 0.32                               | 1500              | 41.3                 |
> | MAGNeT-small   | 3.3                                    | 1.12              | 0.31                           | 180               | 4.0                  |
> | MAGNeT-large   | 4.0                                    | 1.15            | 0.29                          | 180               | 12.6                 |
> | *SoundStorm-small | 7.5                     | 1.63           | 0.18                         | 1518               | 18.1                 |
>
>
> [1] Borsos, Zalán, et al. "Audiolm: a language modeling approach to audio generation." IEEE/ACM Transactions on Audio, Speech, and Language Processing (2023).
>
> [2] Chung, Yu-An, et al. "W2v-bert: Combining contrastive learning and masked language modeling for self-supervised speech pre-training." 2021 IEEE Automatic Speech Recognition and Understanding Workshop (ASRU). IEEE, 2021.

---

### Official Review · Reviewer_Vzfc · 2023-11-01

**Soundness:** 3 good
**Presentation:** 3 good
**Contribution:** 3 good
**Rating:** 6
**Confidence:** 4

**Summary:**

In this paper, the author proposed MAGNET, a masked discrete generative model on top of RQV representations (EnCodec), able to perform audio generation at a faster inference speed than other baselines while reaching the same generation quality. To improve their model, the authors propose a list of techniques: 1. a masking strategy where instead of masking tokens, spans of tokens are masked, given that neighboring tokens are inter-dependent temporally paired with a restricted context on the self-attention for codes greater than one (given the convolutional nature of EnCodec and the limited receptive field of its LSTM in practice) 2. a re-scoring strategy at inference time, where the probability to mask a span is given by an external audio model (e.g., MUSICGEN and AudioGen) 3. an annealer for the classifier free guidance embedding strength. Additionally, the authors experiment with a hybrid model trained both with autoregression and masking. The resulting model results 10 times faster than the autoregressive baseline when using low batch sizes.

**Strengths:**

- The proposed masking strategy, which uses token spans instead of individual tokens, is an important methodological contribution of the paper since it greatly impacts the performance of the proposed model.
- The 10x speedup in latency is remarkable, especially in a setting like music, where it is required to sample many tokens, given the long context.

**Weaknesses:**

- A problem with the work is related to the title and the overall tone, in which the authors claim proposing a new type of audio/music model. In a paper titled "Masked Audio Generative Modeling," it is supposed that the authors proposed for the first time a masked model for audio. Nonetheless, the papers introducing such an idea in audio are SoundStorm (which the authors cite correctly) and VampNet https://arxiv.org/abs/2307.04686 (which the authors should cite as concurrent work). The authors could change the title of the article to better delineate the paper as a set of improvements of such models (providing at least one of the two as baselines).
- Except for the masking strategy, which I find important, the other introduced techniques seem relatively not critical or novel, especially the rescorer model, given that it improves the FAD only marginally.

**Questions:**

- The scheduler $\gamma(i; s) = \cos( \pi(i−1)/ 2s )$ is badly defined because computed on $i=1$ returns still $1$.
- I do not understand why the new CFG anneal performs better.
- Can the authors explain why the small 300M model performs better than the 1.5B model? I find this counterintuitive.
- Additionally, can the authors explain better why the restricted context improves the metrics? Intuitively, the only advantage should be training/inference time improvement.

---

> ### Author Response · Authors · 2023-11-17
> **Official author's response (part 1)**
>
> **Regarding the evaluated baselines:** We would like to first emphasize the main differences between SoundStorm and VampNet to the proposed method.
>
> SoundStrom was proposed to speed up inference time for the AudioLM [1] model, while the authors presented results for the task of text-to-speech only. As it is based on AudioLM, SoundStorm first models semantic tokens in an autoregressive manner, specifically using w2v-bert tokens [2]. Hence, it is not clear how to convert such a model to text-to-music generation considering the semantic tokens needed for the task. Nevertheless, following the reviewer's request, we trained a version of SoundStorm, using the exact setup as described in the original soundstorm paper while we consider the first EnCodec codebook as semantic tokens. Results are reported below. As can be seen MAGNeT performs better across all metrics. Notice, the number of steps of the SoundStorm model is significantly higher than MAGNeT due to its autoregressive modeling of the first codebook.
>
> As for the VampNet model, the authors presented results for music inpainting and super-resolution and did not explore full-blown text-to-music generation without an additional **audio-prompting**. Moreover, unlike the proposed method, VampNet is based on two different models, one for the "coarse” tokens and one for the “fine” tokens. MAGNeT uses a single model to generate all tokens. We updated the manuscript to include VampNet in the related work.
>
> |Model  | Fad$_{vgg} \downarrow$ | Kl $\downarrow$ | Clap$_{scr} \uparrow$ | # Steps | Latency (s) |
> |-----------------|----------------------------------------|--------------------------|---------------------------------------|--------------------------|--------------------------|
> | Mousai          | 7.5                                    | 1.59               | 0.23                            | 200               | 44.0                 |
> | MusicLM         | 4.0                                    | -                        | -                                     | -                 | -                    |
> | AudioLDM 2      | 3.1                                    | 1.20               | 0.31                                    | 208               | 18.1                 |
> | MusicGen-small | 3.1                                    | 1.29               | 0.31                              | 1500              | 17.6                 |
> | Musicgen-large | 3.4                                    | 1.23               | 0.32                               | 1500              | 41.3                 |
> | MAGNeT-small   | 3.3                                    | 1.12              | 0.31                           | 180               | 4.0                  |
> | MAGNeT-large   | 4.0                                    | 1.15            | 0.29                          | 180               | 12.6                 |
> | *SoundStorm-small | 7.5                     | 1.63           | 0.18                         | 1518               | 18.1                 |
>
>
> **Regarding the novelty of the proposed method:** We would like to state our contribution: (i) We present a **single** non-autoregressive model for the task of audio modeling and generation (this is in contrast to prior work which uses a cascade of models). We empirically demonstrate the proposed method is able to generate relatively long sequences, **30 seconds long**; (ii) We leverage an external pre-trained model during inference to improve generation quality via a rescoring method; and (iii) We show how the proposed method can be combined with autoregressive modeling to reach a single model that performs joint optimization (Hybrid-MAGNeT).
>
> **Regarding the impact of the rescorer:** In Table 3, we presented an ablation study to better understand the effect of the rescorer. Results suggest that the rescoring method does improve model performance across all metrics, both objective and (more importantly) human study.
>
> Moreover, we believe the usage of external models to guide and constrain the generation process (e.g., the rescoring method), especially in the non-autoregressive setup, opens up a new and exciting research direction that we hope the community will find interesting and valuable.

---

> ### Author Response · Authors · 2023-11-17
> **Official author's response (part 2)**
>
> **Regarding the impact and motivation of the restricted context:** The EnCodec model was optimized using residual vector quantization. Meaning that codebooks greater than one, are mainly influenced by the previous codebooks at the same time-step. Following that modeling paradigm, MAGNeT should not learn dependencies between the different timesteps outside the EnCodec receptive field window for codebooks above one.
>
> In Table 2, we presented an ablation study of the proposed method with and without a restricted context constraint. Results suggest that restricting the context of codebooks higher than 1 constantly improves results. As stated before, this method highlights a specific modeling approach designed for the chosen encoding method. From listening tests, we additionally observe that when we do not follow the restricted context method, the model discards the harmony/music texture and mainly focuses on rhythmic patterns. We updated the demo page on the supplemental material with several music samples demonstrating such an effect.
>
> **Regarding the scheduler:** We believe there is a misunderstanding as at the beginning of iterative decoding we would like to have a 100% masking rate. Which is exactly what we get when setting i=1. We would be happy to provide more details if we misunderstand the reviewer.
>
> **Regarding CFG annealing ablation:** We follow a similar approach to the one proposed by [3] for textually guided 3D shape generation. In preliminary studies, we found such an approach to perform better than a CFG without annealing.
>
> In the Table below we provide results of the proposed method considering different settings of CFG. We report FAD, KL, and CLAPScores. Results suggest that while some settings provide comparable performance considering one metric, following CFG annealing using 10->1 provides the best results considering all metrics.
>
>
> $\lambda_0 \rightarrow \lambda_1$      | 1 $\rightarrow$ 1 | 3 $\rightarrow$ 3 | 10 $\rightarrow$ 10 | 10 $\rightarrow$ 1 | 20 $\rightarrow$ 20 | 20 $\rightarrow$ 1 |
> |----------------------------------------|-------------------|-------------------|---------------------|--------------------|---------------------|--------------------|
> | Fad$_{\text{vgg}} \downarrow$ |  3.95  | 0.99  | 0.63  | **0.61**   | 0.80   | 0.68 |
> | Kl $\downarrow$               | 0.79 | 0.60 | 0.56   | 0.56  | 0.57   | 0.56 |
> | Clap$_{\text{scr}} \uparrow$   | 0.13 | 0.28 | 0.28   | 0.31  | 0.31   | 0.31 |

---

> > ### Author Response · Authors · 2023-11-17
> > **Official author's response (part 3)**
> >
> > **Regarding larger model perform worse than smaller ones:** This is noticeable especially when considering the FAD metric, and was also observed in the original MusicGen paper, see [4], Table 1. Notice that it is not the case for human study.
> >
> >
> > **Regarding the title of the paper:** Thanks for highlighting the potential gaps in the name of our paper. We will consider changing it.
> >
> >
> >
> > [1] Borsos, Zalán, et al. "Audiolm: a language modeling approach to audio generation." IEEE/ACM Transactions on Audio, Speech, and Language Processing (2023).
> >
> > [2] Chung, Yu-An, et al. "W2v-bert: Combining contrastive learning and masked language modeling for self-supervised speech pre-training." 2021 IEEE Automatic Speech Recognition and Understanding Workshop (ASRU). IEEE, 2021.
> >
> > [3] Sanghi, Aditya, et al. "CLIP-Sculptor: Zero-Shot Generation of High-Fidelity and Diverse Shapes From Natural Language." Proceedings of the IEEE/CVF Conference on Computer Vision and Pattern Recognition. 2023.
> >
> > [4] Copet, Jade, et al. "Simple and Controllable Music Generation." arXiv preprint arXiv:2306.05284 (2023).

---

> ### Comment · Reviewer_Vzfc · 2023-11-22
>
> I thank the authors for the additional studies and the clarifying comments, especially regarding the restricted context, now I understand the idea and I find it important. Regarding the title of the paper, while it is true that VampNet does not perform full text-to-audio, nonetheless, it performs "masked audio generative modeling." If the authors propose a new title that better differentiates their contribution from VampNet, I will increase my score.

---

> > ### Author Response · Authors · 2023-11-22
> > **Official author's response**
> >
> > We thank the reviewer for the quick response. Following the reviewer's suggestion, we will change the paper title to be: Masked Audio Generation using a Single Non-Autoregressive Transformer.

---

> > > ### Comment · Reviewer_Vzfc · 2023-11-22
> > >
> > > I thank the authors for proposing a new title, which better fits the contribution of the paper. I have raised my score accordingly.

---

### Author Response · Authors · 2023-11-17
**Comment to all reviewers**

We would like to thank the reviewers for taking the time to review our manuscript and provide meaningful feedback.

Below we address each of the points raised by the reviewers. We will be happy to provide additional clarification to any remaining concerns.

---

### Author Response · Authors · 2023-11-21
**Followup discussion with the reviewers**

As we approach the authors-reviewers discussion deadline, we would like to kindly ask if the reviewers have any additional questions.

We would be happy to provide additional clarifications for any remaining concerns.

---

### Meta-Review · Area_Chair_E4v6 · 2023-12-04

**Metareview:**

This paper presents MAGNET, a new method for audio sequence modeling that uses a single-stage, non-autoregressive transformer. It enhances audio quality through a rescoring method and a hybrid model that combines autoregressive and non-autoregressive models. It’s efficient for text-to-music and text-to-audio generation, comparable to baselines but seven times faster. It also highlights the trade-offs between different modeling types considering latency, throughput, and quality.

The authors have successfully addressed the questions posed by the reviewers in their rebuttal, leading to an increase in the score from one of the reviewers. In the end, all reviewers consistently recommended acceptance of the paper.

**Justification For Why Not Higher Score:**

The paper’s contributions do not appear to be significant enough, and the methods proposed lack sufficient novelty for a higher score.

**Justification For Why Not Lower Score:**

The paper is well-written and offers clear and comprehensive experimental validation.

---

### Decision · Program_Chairs · 2024-01-16

Accept (poster)